# Development of Mixed Flow Fans with Bio-Inspired Grooves

**DOI:** 10.3390/biomimetics4040072

**Published:** 2019-10-18

**Authors:** Jinxin Wang, Toshiyuki Nakata, Hao Liu

**Affiliations:** 1School of Naval Architecture, Ocean and Civil Engineering, Shanghai Jiao Tong University, Shanghai 200240, China; rafewang@sjtu.edu.cn; 2Graduate School of Science and Engineering, Chiba University, Chiba 263-8522, Japan; tnakata@chiba-u.jp; 3Shanghai Jiao Tong University and Chiba University International Cooperative Research Center, Dongchuan Road 800, Shanghai 200240, China

**Keywords:** bio-inspired grooves, computational fluid dynamics, mixed flow fan, turbulence kinetic energy, groove forms, groove design parameter exploration

## Abstract

Mixed flow fan is a kind of widely used turbomachine, which has faced problems of further performance improvement in traditional design methods in recent decades. Inspired by the microgrooves such as riblets and denticles on bird feathers and shark skins, we here propose biomimetic designs of various blades with the bio-inspired grooves, aiming at the improvement of the aeroacoustic performance. Based on a systematic study with computational fluid dynamic analyses, we found that these designs had the potential in noise suppression even with macroscopic grooves. Our best design can suppress turbulence kinetic energy by approximately 38% at the blade leading edge with aerodynamic efficiency loss of only 0.3 percentage points. This improvement is achieved by passive flow control. The vortical structures are changed in a favorable way at the leading edge due to the grooves. We believe that these biomimetic designs could provide a promising future of enhancing the performance of mixed flow fans by making grooves of ideal flow passages on the suction faces of blades in accord with the theory of pump design.

## 1. Introduction

Mixed flow fan is widely used for the ventilation in industries and it is a kind of centrifugal pump or turbomachine which has been developed for centuries [1]. Its specific speed ranges from 300 to 800, which covers the range between the radial pump and the axial flow pump, resulting in a higher pressure output within a broader range of flow rate [2]. The specific speed is defined as below:(1)$$n_{s}=\frac{3.65n\sqrt{Q}}{H_{pump}^{3/4}}$$
where, the n denotes rotational speed (rpm), Q denotes the volume flow rate (m^3^/s) and $H_{pump}$ (m) denotes the head of the pump (the dimension of $n_{s}$ is same to $g^{3/4}$ (g: gravity acceleration)) [2]. In general, the industrial flow fans require improvement on the fluid dynamic performance and the working stability associated with the vibrations, noise, and unnecessary forces [3]. To this end, recent studies on the industrial pump mainly focus on the effects of its traditional design parameters on the fluid dynamic output or pressure fluctuations [4,5,6,7,8]. For example, the spanwise twist change may increase efficiency by 0.8% but cause larger pressure fluctuation amplitude [5]. The tip clearance (TC) variations even within 0.5 mm can result in significant changes in efficiency and pressure fluctuation [7]. As a result of the previous studies, the performance of the current mixed-flow fans is maximized in the traditional design space. A novel design principle is, therefore, necessary for further improving the performance.

To improve the performance of turbomachines, many researchers adopted the passive flow control. Passive control techniques are those that require no auxiliary power and no control loop [9,10,11,12]. Volino et al. investigated the function of a bar or some cylinders near the leading edge of a turbine airfoil. They found that these objects can control the boundary layer transition and reattachment [13,14]. McAuliffe and Yaras successfully manipulated the breakdown of a separation bubble by making various surface modifications [15]. On the other hand, animals also perform the passive flow control during their motions mainly through structural and morphological components of the body [16], like serrations and tubercles [17,18,19]. Nowadays, people have been increasingly realizing the value of strategies animals adopt. These biomimetic ideas have been inspiring researchers and are already applied on airfoils, propellers and even rudders [20,21,22].

The kinematics of biological wings or fins share similarities with simple rotations of turbomachines, i.e., their fluid mechanical boundary conditions are similar. Thus, it is possible that the biomimetic method can reproduce the excellent performance on turbomachines. The recent decade has already witnessed such successful biomimetic designs [23,24,25]. It is known that the surface of animals, such as shark skins or bird feathers, are formed by multiscale groove structures [26,27,28] that can reduce the drag by passively modifying turbulence conditions over the surfaces [27,29]. Therefore, it is of great importance to modify the blade surface flow by applying bio-inspired grooves on mixed flow fans. Recently, grooves or other similar structures start to be introduced to pump designs [30,31]. Especially, the impeller with “bionic three-dimensional (3D) fluting” named ZAbluefin adopts the so-called “rippled blade shape”, which is actually a grooved blade. The company claimed this design can reduce the sound level by diffusing sound radiation significantly and increase efficiency by delaying separation [31].

Therefore, aiming at exploring a biomimetic fan design that can achieve better aeroacoustics performance through passive flow control, we carried out a systematic computational fluid dynamic (CFD) analysis to investigate the effects of bio-inspired grooves over the impeller surface of a mixed flow fan on the aerodynamic and the aeroacoustics performance. The computational models were validated indirectly through the comparison with our previous experimental results. The performance was evaluated by total pressure efficiency and turbulence kinetic energy and the design parameters were explored to maximize the performance. We then demonstrate that the proposed fan design does show potential in suppressing noise though with marginal aerodynamic loss, and that the biomimetic design could provide a useful and effective method to improve the aeroacoustic performance of mixed flow fans.

## 2. Materials and Methods

### 2.1. The Original Impeller and the Computational Fluid Dynamic (CFD) Modelling 

In this study, we took the previous mixed flow fan ([32] ALF-528S, Teral) as the research object. Figure 1a shows the picture of the original impeller which we call the basic design. Table 1 gives more details on this mixed flow fan and the operating condition in our study which is around the design condition of this fan.

Figure 1b shows the numerical model we adopted to do simulations where only one blade was analyzed with the periodic boundary conditions (BC). We also included the duct and the casing in this study. We did not get the steady vanes involved because we want to mainly focus on the effects of different blade designs without the fluid field being affected by them. Therefore, aerodynamic performance derived from this model is higher than the reality since this highly simplified model removes motor chamber and steady vanes which cause friction and impact loss. 

The boundary condition at the inlet is set to Opening with the relative pressure of 0 Pa and 5% turbulence intensity while a periodic mass flow rate (1/6), as given in Table 1, is fixed at outlet. These settings can reflect the reality since we let the blades create negative pressure by rotating to suck air from the atmosphere while controlling the outlet flow rate by a damper. In addition, our models contain two domains of a static duct domain and a dynamic rotational domain. These two domains are smoothly connected through a technique of the so-called general grid interface (GGI) in the multiple frame of reference (MFR) and the frame change model is set to Frozen Rotor (to fix the rotor). The periodic boundary condition is applied to both domains so as to reduce the computation cost. The Reynolds number at the operating condition, as in Table 1, is about 446,000.

We adopted the incompressible and steady RANS (Reynolds Averaged Navier–Stokes) equations with the SST (Shear Stress Transport [33]) turbulence model to solve this problem. The governing equation can be written as:(2)$$\frac{\partial \bar{u_{j}}}{\partial x_{j}}=0$$
(3)$$\bar{u_{j}}\frac{\partial \bar{u_{i}}}{\partial x_{j}}=-\frac{1}{\rho }\frac{\partial \bar{p}}{\partial x_{i}}+\frac{\mu }{\rho }\frac{\partial^{2}\bar{u_{i}}}{\partial x_{j}\partial x_{j}}-\frac{\partial }{\partial x_{j}}\left(\bar{u\prime_{i}u\prime_{j}}\right)$$
where, ρ denotes the density, $\bar{p}$ denotes the mean pressure, μ denotes the dynamic viscosity and $\bar{u}$ is the mean velocity. The equations of SST turbulence model developed by Menter are as follows:(4)$$\frac{\partial \left(\rho U_{i}k\right)}{\partial x_{i}}=\tilde{P_{k}}-\beta^{*}\rho k\omega +\frac{\partial }{\partial x_{i}}\left[\left(\mu +\sigma_{k}\mu_{t}\right)\frac{\partial k}{\partial x_{i}}\right]$$
(5)$$\frac{\partial \left(\rho U_{i}\omega \right)}{\partial x_{i}}=\alpha \rho S^{2}-\beta \rho \omega^{2}+\frac{\partial }{\partial x_{i}}\left[\left(\mu +\sigma_{\omega }\mu_{t}\right)\frac{\partial \omega }{\partial x_{i}}\right]+2\left(1-F_{1}\right)\rho \sigma_{\omega 2}\frac{1}{\omega }\frac{\partial k}{\partial x_{i}}\frac{\partial \omega }{\partial x_{i}}$$
where, $U_{i}$ is the time-averaged velocity and ω is the turbulent frequency. k is the turbulence kinetic energy (TKE) which will be later utilized for the aeroacoustic performance evaluation in Section 2.3. $F_{1}$ is the blending function, S is the invariant measure of the strain rate, $\beta^{*}$ is 0.09, and $\sigma_{\omega 2}$ is 0.856. The production limiter $\tilde{P_{k}}$ is used to prevent the buildup of turbulence in stagnant regions. As the commercial software ANSYS CFX 14.5 was used for all analyses, other constants’ values in the above equations can be found in Reference [34], which are derived from a blend of the corresponding constants for the k−ω and k−ε models. The High-Resolution scheme was adopted for advection terms in both continuity (1) and momentum equations (2), which is a blending scheme between the first order upwind differencing scheme and the second-order central differencing scheme. We also utilized this scheme to solve the turbulent flow. In this study, we focus on the turbulent flow near the leading edge of the blade, which we will later clarify in detail in Section 2.3. Since the Reynolds number in the present study was calculated to be on the order of $10^{5}$, the turbulence model was employed. We chose the steady RANS model here to get a primary understanding of the turbulent flow due to the limitations of our computational resources. The SST model is recommended for high-accuracy boundary layer simulations [35]. According to Reference [35], to benefit from this model, a resolution of the boundary layer of more than 10 points is required. Results derived from the SST turbulence model with different sets of mesh will be discussed in detail in Section 2.4.

We set RMS (Root Mean Square) residual < $2\times 10^{-6}$ as the convergence criteria, which is also applied to the turbulence equation solving. The adopted criteria should be accurate enough as the CFX recommends $1\times 10^{-5}$ for common problems of research level. Cases that cannot be converged in our analyses will be ensured to have outlet pressure fluctuating below 0.5% within the last 300 false timesteps. The Physical Timescale of the false timestep we utilized in our analyses is $1/\omega_{r}$ ($\omega_{r}$ (denotes the rotational speed), which is also recommended by CFX [35].

### 2.2. Biomimetic Designs and the Design Method

Figure 2 shows different groove forms we explored in our research. Here, we proposed two kinds of grooves, the wavy shape, and the riblet shape. The wavy shape is similar to the “sawtooth” shape in Reference [29] but affects both the suction and pressure face side. The riblet shape is exactly the “blade riblet” mentioned in Reference [29] but we here merely call it “riblet shape” to avoid possible misunderstandings caused by impellers’ “blade” often used in pump designs. We explored the riblet shape in two forms, on the suction face and the pressure face, since the wavy shape also results in grooves on the pressure face side, which means there are three biomimetic blade forms in total.

Figure 3 provides the illustration of the biomimetic design method in detail. We first divided the blade averagely into several paths in the radial direction. For example, there are 10 paths in Figure 3. According to the pump design theory [2], ideal flows on these paths do not interact with each other. Then, we built the grooves in different forms based on these paths with each groove occupying two. Thus, the grooves aligned with these paths cause no fluid impact loss theoretically and the groove number *N* is defined by the half of the path number. Researchers take h/s as an important geometry feature of biomimetic grooves and the range between 0.1 and 1 is often investigated [29,36]. We design grooves by keeping their cross-section area constant at the same *N* value while ensuring our designs’ h/s drops within this range as well. The height of the wave peak or valley is always 3 mm, as shown in Figure 3a, which makes the h/s (of different *N*) ranges from 0.3 to 0.6 at the leading edge. For riblet forms, we need to add a Δℎ to keep their groove cross-sections as the same areas to the wavy forms shown by the shade in Figure 3 for considering the thickness effect and the area varying slightly with different *N* at the wavy blade root and tip. Hence, the riblet heights are 3.8 mm, 4.0 mm, and 4.2 mm for groove number 5, 7 and 9 respectively, as shown in Table 2, which makes the *h/s* 0.2, 0.3 and 0.4 at the leading edge for riblet shape designs.

For further study, we also need to change the height and length of the riblets. Thus, height is explored in three values, 4.2 mm, 5.9 mm, and 7.6 mm when *N* is 9, as shown in Table 2. The height of 4.2 mm ensures riblet grooves (*N* = 9) of the same cross-section area with the wavy grooves (*N* = 9), while the latter two values make the area increase to 1.5 and 2 times (h/s = 0.60, 0.8). Also, the length will be explored in the other two values which are 1/3 L and 2/3 L. L denotes the original rib length from the leading edge to the trailing edge, as in Figure 2f.

### 2.3. Evaluation Method of Aerodynamic and Aeroacoustic Performance

In this study, to evaluate the designs’ aerodynamic performance, we check the total pressure efficiency of each design as bellow:(6)$$\eta =\frac{\left(P_{out}-P_{in}\right)\cdot Q}{\tau \cdot \omega },$$
where, P, Q, τ and ω are the total pressure, volumetric flow rate, torque, and angular velocity, respectively. The subscripts “in” and “out” represent the inlet and outlet, respectively.

To evaluate the designs’ aeroacoustic performance, we check the TKE on a Spanwise Face (SF) and an Offset face from the Suction face (Off_Suc), as shown in Figure 4. The spanwise face is a rotational sweeping face around the impeller’s axis by a line on the blade 5 mm away from the leading edge in the chordwise while the offset face is a face 1 mm from the suction face in its normal direction. Hence, they are dubbed SF_5mm and Off_Suc_1mm, respectively. We will take advantage of the TKE area integrals on SF_5mm to estimate the potential noise reduction ability of different designs. TKE is defined as the variance of the fluctuations in velocity [34,37], which is known as k in the k-ω-based SST turbulence model (Equations (4) and (5)) and can be written as:(7)$$k=\frac{1}{2}\bar{u_{j}^{\prime }u_{j}^{\prime }}$$

We believe it is reasonable to evaluate the noise level indirectly by checking the TKE. Results in Reference [32] of the tuft method show the strong turbulence at the blade’s leading edge on the suction surface, implying a strong broadband noise source. Moreover, both the tuft method (in Reference [32]) and the TKE check method (which will be presented later in this section) confirmed that the serrations reduce the turbulence near the blade leading edge greatly when the specific noise level tested in experiments was reduced by 1.5 dB. In addition, some researchers also take advantage of the TKE derived from steady or unsteady RANS analyses to do noise prediction [38,39,40,41].

The flow over the blade surface can be taken as the fully turbulent flow [42,43]. We took a flow case of NACA 0012 whose Reynolds number is 500,000 in Reference [43] as a reference since its Reynolds number is close to ours. For the NACA 0012 airfoil, transition happens quite near the leading edge when the Reynolds number is 500,000 and the angle of attack is over 8 degrees. Moreover, this location will move upstream when the NACA airfoil’s maximum thickness tends to be smaller [43]. We found the incidence angle of the blade at our operating condition is over 8 degrees from the middle to the tip of the leading edge (the incidence angle here can be seen as the angle of attack and it can be calculated through the method in Reference [2]). Besides, the blade’s thickness (1.65 mm) is considerably lower than 12% of the chord length (NACA 0012). Hence, the flow condition over the blade is fully turbulent, and the transition happens pretty close to the leading edge. Therefore, we chose SF_5 mm because it is not only quite close to the leading edge but also far enough to turn the flow into a fully developed turbulent one [42,43]. Furthermore, Off_Suc_1 mm is chosen because it locates not only distantly enough to the viscous sublayer (y+ = 100 > 30, y+ is based on the estimated Reynolds number of 446,000) according to Reference [37], but also within the prism layers we meshed. Thus, we can obtain good views of TKE conditions in the turbulent layers with a high resolution by checking contours of different designs on Off_Suc_1 mm.

Here, we would like to validate the prediction of the TKE reduction by the single blade model indirectly with the help of some results in our previous study. We first validate a 2-blade full size model with the experimental results. Figure 5a shows the experimental set-up in Reference [32]. The experimental method conformed on JIS (Japan Industrial Standard) B8330 is described in detail here. The mixed flow fans were connected to ducts on both sides. The flow straightener (grid and wire net) were installed at the outlet duct. The input power is measured by a power meter and a current clamp-on probe (WT333E and 96001, Yokogawa Test and Measurement Corporation). The static and dynamic pressures were measured at the outlet duct by a manometer (DPC-201N12, OKANO WORKS, LTD.), and a pitot tube and a manometer (LK-1S and DMC-102N11, OKANO WORKS, LTD.), respectively. The air density is measured by a combined pressure, humidity and temperature transmitter (PTU303, Vaisala). Figure 5b,c shows the whole picture and the local details respectively of the 2-blade full-size model which contains three domains. Note that the steady vanes, shaft and motor chamber are included in this model for validation. It is also a kind of periodic numerical model, but it keeps as many real boundary conditions as possible and the blade interaction effects can be involved and compared with the single blade model. The disadvantage of it is that this model’s computational cost is much higher. The key mesh parameters adopted for this numerical model are same to the Mesh Set 2 (see Section 2.4). In experiments, the straightening grid and wire net illustrated in Figure 5a could make the air flow in the duct more axially, which reduces the friction loss by the duct wall due to the circumferential velocity. Thus, the aerodynamic performance derived through CFD analyses are lower than the experimental results, as shown in Figure 5d,e, since we left out the hard modelling of the straightening grid and wire net in our numerical models. Therefore, from Figure 5d,e, we confirmed that the 2-blade full size model has been validated. 

The TKE results of the basic and serration design by the 2-blade full-size model are shown in Figure 6a,b, while the results under the same operating condition respectively, by the single blade model are shown in Figure 6c,d. Results in Figure 5d,e and Figure 6a,b of two kinds of designs (basic and serration designs) suggest that the serrations can reduce the TKE near the leading edge while not affecting the aerodynamic performance. More explorations of different serrations’ designs and conclusions are in Reference [32]. Here, it is more important that results in Figure 6 of the two numerical models (different in the blade number) remind us that the interaction between blades will considerately increase the TKE value on SF_5 mm. However, the average reduction of the TKE area integrals on SF_5 mm by serrations of each blade is still quite close. Average reduction by each blade derived from the 2-blade full-size model is 0.00615 m^4^/s^2^, while the value derived from the single blade model is 0.00584 m^4^/s^2^, which is about 5% lower than the former. Besides, we can also see the obvious TKE reduction near the leading edge from both numerical models, as shown in black circles in Figure 6. Therefore, we believe that it is reasonable to firstly utilize the single blade model in capturing the key flow features of the biomimetic design in terms of the TKE reduction. Furthermore, this model can save our computing time and let us try different biomimetic designs as many times as possible.

### 2.4. Numerical Grid

The grid systems we adopted for different designs are shown in Figure 7. All blades are refined with the size of 0.7 mm at the leading-edge region, as shown in Figure 7a. This refined region covers 10 mm from the leading edge in the chordwise direction. We impose 11 prism layers on all the blade surfaces and the casing walls with a growth rate of 1.06. Other parameters of this grid system can be checked in the Mesh Set 2 column provided by Table 3. According to Reference [35], the best near-wall grid resolution is to make at least y+ < 2, but CFX provides the Automatic near-wall treatment that allows for a smooth shift from a low-Reynolds number form to a wall function formulation. To take advantage of the reduction in errors offered by the automatic switch, users should resolve the boundary layer using at least 10 nodes when adopting the SST turbulence model [35]. Hence, here we only impose 11 layers around the blade when y+ is 10 (first layer height: $1\times 10^{-4}$ m) for the prism layer thickness, which is limited by the clearance between the blade tip and the casing wall and we want to save computing costs as well. In addition, we checked the maximum y+ on the blade surface in the Mesh Set 2 at the operating conditions in Table 1, which is below 15.

We also did the grid independence verification by exploring another two sets of mesh in Table 3. We carried out the CFD analyses of the basic design at the operating condition in Table 1. The aerodynamic and aeroacoustic performance by the three sets in Table 3 and the TKE contour shown by Figure 8 suggest that the Mesh Set 2 can produce quite close results to the finest one. Therefore, we adopted the Mesh Set 2 for our study.

## 3. Results

### 3.1. Groove Form Effects

The effects of three groove forms were investigated, which are the wavy form, the riblets on the suction face form and the riblets on the pressure face form. We assigned three different groove numbers for each groove form to determine which form is the best at the TKE suppressing while not causing too much aerodynamic loss. Figure 9 gives the results of different biomimetic design sets and some other varied suction riblets designs. It also includes results of the basic and the previous serrations design. The ordinate denotes the total pressure efficiency and the abscissa denotes the integrated TKE value on the SF_5 mm. Hence, the top left corner represents the best result of solving the tradeoff between aerodynamic and aeroacoustic performance while the bottom right corner represents the worst result. 

As Figure 9 shows, it is noticeable that our previous design, the serrations design, outperforms other designs. Among the designs we study this time, the wavy shape forms can reduce the most TKE near the leading edge but also cause the most aerodynamic loss. We made Figure 10, TKE contours on the Off_Suc_1 mm as in Figure 8, to check the whole blade leading-edge region. Figure 10 also shows highly suppressed TKE by the wavy shape design. Figure 9a indicates that riblets on the suction face design set can achieve a better balance between the aerodynamic loss and the TKE reduction than the other two forms, although their ability to suppress turbulence is slightly lower than the wavy shape designs, as shown in Figure 10. The riblets on the pressure face design performs worst. This is not so hard to expect because the turbulence mainly happens on suction faces instead of pressure faces and its TKE contour seen in Figure 10d is almost the same as the basic design.

We think the riblets on the suction face form is the best biomimetic form in this study as it is closest to the top left corner. Therefore, we picked this form to continue our exploration. Furthermore, in this form set, the total pressure efficiency of nine riblets design is only 0.1 percentage point lower than the five riblets, one while the former’s turbulence suppressing amount is about 21 percentage points higher than the latter. Hence, our following explorations are based on the design “Rib N9 H4.2”.

### 3.2. Groove Parameter Effects

It is clear that the groove number plays an important role in turbulence suppressing for the wavy and riblets on the suction face form, as seen from Figure 9a, while all the efficiency slightly varies (within 0.5%) within each form set. However, as for the riblets on the pressure face form, increasing groove number results in little effect on the TKE reduction and their efficiency varies within 1%. 

We continued to investigate the effects of groove parameters by changing groove height and length based on the riblet shape form of nine grooves. As Figure 9b shows, the increase of the height from 4.2 mm to 7.6 mm causes more loss on aerodynamics and lower ability of TKE reduction. Here, we did not explore designs with the height value below 4.2 mm because it is hard to impose 11 prism layers on them. We thus continue to explore the riblets length effect still based on the design “Rib N9 4.2”.

Figure 9b suggests that points representing designs of different riblets’ length cluster together in a black circle, implying that their performance is close. Therefore, the length is not so important if it is long enough to cover the leading-edge region. As seen from Figure 11, 1/3 ribs’ original length is enough to keep the same TKE reduction effect at the leading-edge region.

## 4. Discussion

We determined that the riblets on the suction face design is the best form among all designs given the same groove cross-section area due to its ability to better solve the tradeoff problem between aerodynamic and aeroacoustic performance. Furthermore, we can take the “Rib N9 H4.2” as the best design in this form. It can reduce the TKE by about 38% with a corresponding efficiency sacrifice of 0.3 percentage points. Besides, the results of another riblet shape form suggest that riblets on the pressure face can hardly affect the turbulence over the suction face.

We think the groove number is a critical parameter in groove designs because they directly affect the riblets’ function of stabilizing the flow while the ribs’ length is less important if they can cover the leading-edge region. Besides, it is unnecessary to build high riblets since larger height could harm the riblets’ function. Results showing that larger groove number results in better TKE reduction, we think are to some extent similar to the serrations’ feature [18], the denser the grooves or serrations are, the ability to passively stabilize the flow on the surface is better.

The bio-inspired grooves can considerably change the flow structures near the leading edge in a favorable way. Figure 12 provides the velocity vectors projected on the Off_set_1 mm surface in the rotational frame. It can be obviously found that the flows above different designs’ suction surface were totally different. As shown in Figure 12a,b, on the basic blade suction surface, the fluid near the leading-edge flows from two sides to each other, which are the blade root side and the tip side. Flows from the two sides meet at some place closer to the blade tip and then move towards the trailing edge, which results from the joint effect of the centrifugal force and impacts back from the casing wall. However, the bio-inspired grooves can block these motions and enforce the air to flow in the narrow passages formed by them, which is more similar to the ideal flow in the pump theory [2]. What is more important is that we found that these grooves are able to result in the vortices near the leading edge, as shown in black circles in Figure 12c,d, which definitely own a component in the axial direction (or Z direction in our simulations). Thus, we continue to investigate the vortical structures by checking vorticity in the Z direction for convenience in the software CFD-Post.

Figure 13 provides us with the axial vorticity on the Off_Suc_1 mm and vortex cores (Q = 0.02) for the basic and other two successful biomimetic designs. It is apparent that more axial vorticity has been produced by the grooves compared with the basic design seen from Figure 13. This is quite similar to the phenomena described in References [17,18,19], where small vortices or eddies are generated by biomimetic structures near or at the leading edge to realize the passive flow control. In Reference [17], the small vortices which the leading edge vortex is broken up into by serrations are capable of mitigating the transition from laminar to turbulence downstream, and in Reference [19], the tubercles can generate a pair of symmetrical vortices in inverse directions near the leading edge to suppress the separation downstream. Therefore, the vorticity generated here probably plays an important role of stabilizing the flow near the leading edge. As shown in Figure 13b,d,f, the leading-edge vortex in our cases is also broken up into smaller ones by grooves. We believe this is the reason why more grooves can result in more TKE reduction because more vortices favorable in the passive flow control are produced.

It is interesting that the vortical structures generated by the two biomimetic designs are different. From Figure 13c, d, we can easily tell the vorticity between the riblets are almost in the positive axial direction while the wavy shape grooves create a pair of asymmetrical vortices in inverse directions. The small axial vorticity actually results from the impacts on the grooves’ walls, as shown in Figure 12c,d, and probably the surface-tilted angle formed by the wavy shape has an important effect on the formation of the negative axial vorticity. This negative vorticity should also be beneficial to the TKE suppressing and perhaps is the reason why the wavy shape can better suppress the TKE at the leading-edge region, which can be seen from Figure 10b,c. However, the wavy forms also increase the blade areas on both the pressure and the suction face, which causes more fluid contact area than the riblet forms and thus, induces more friction loss. 

## 5. Conclusions

In this study, we explored three bio-inspired groove forms on a mixed flow fan designed in accordance with the ideal flow assumption in pump theory. We also explored the effects of different design parameters of the riblet shape. While the experimental analyses need to be carried out in order to investigate the noise reduction effect by these biomimetic designs further, here are the findings we get through the systematic simulations:The wavy shape form is better at reducing the TKE associated with the broadband noise than the riblet ones with the same groove cross-section area. However, the riblets on the suction face form outperforms others by excellently solving the tradeoff problem between aerodynamic loss and TKE reduction. Our best design can suppress the turbulence kinetic energy by approximately 38% at the blade leading edge, while its aerodynamic efficiency loss is merely 0.3 percentage points.More grooves can result in more TKE reduction at the leading-edge region. Large groove height may harm grooves’ TKE reduction ability. For the riblet design, the riblets’ length does not play a critical role in both aerodynamics and aeroacoustics as long as the length is enough to cover the leading-edge region.Bio-inspired grooves can break the leading-edge vortex up into smaller vortices or eddies and result in higher vorticity concentrated at this region. This passive flow control near the leading edge probably suppresses the TKE associated with the broadband noise successfully.

## Figures and Tables

**Figure 1 biomimetics-04-00072-f001:**
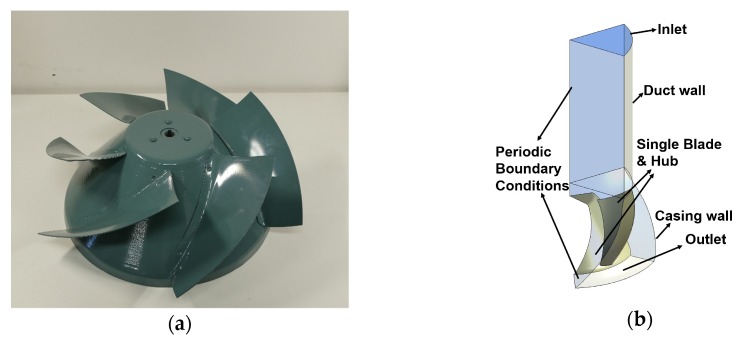
The basic design: (**a**) Original impeller, (**b**) Single blade numerical model.

**Figure 2 biomimetics-04-00072-f002:**
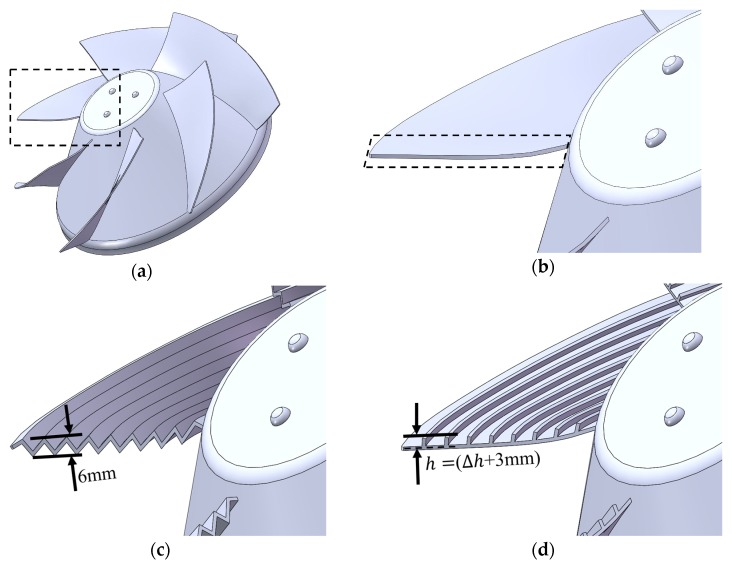
Basic and biomimetic blade designs, groove number *N* = 9: (**a**) Whole basic impeller, (**b**) Basic shape, (**c**) Wavy shape, (**d**) Riblet shape_ Suction face, (**e**) Riblet shape_ Pressure face, and (**f**) Riblet shape_ Suction face_L1/3. The rectangle region in (**a**) is exaggerated as in other figures. The lozenge region in (**b**) is the leading-edge region.

**Figure 3 biomimetics-04-00072-f003:**
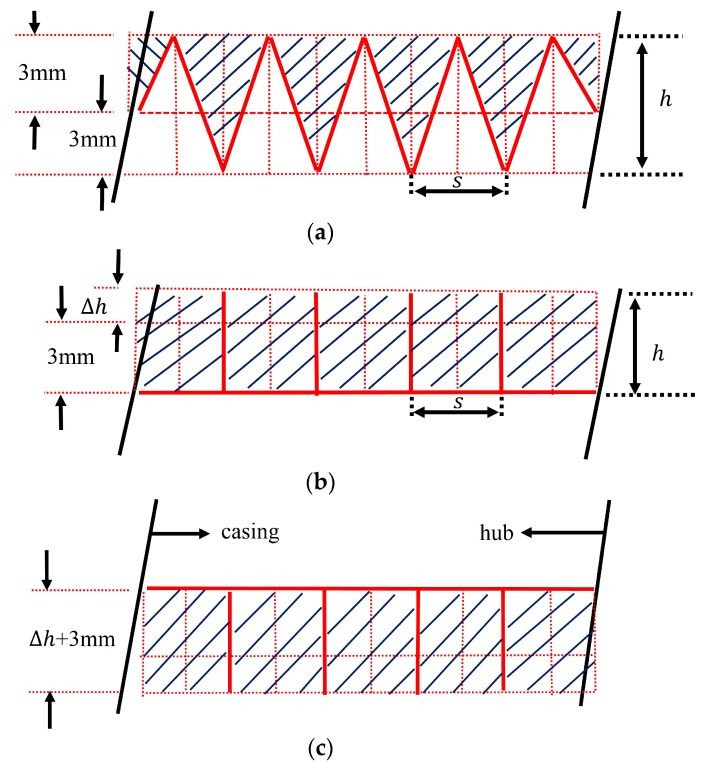
Grooves design illustration, groove number *N* = 5: (**a**) Wavy shape, (**b**) Riblet shape_ Suction face, (**c**) Riblet shape_ Pressure face. The solid red lines are the blade spanwise sections and the dashed red lines the divide blade cross-section into paths (there are 10 paths). The shade areas of the same *N* are constant regardless of blade forms which denote the groove cross-section area.

**Figure 4 biomimetics-04-00072-f004:**
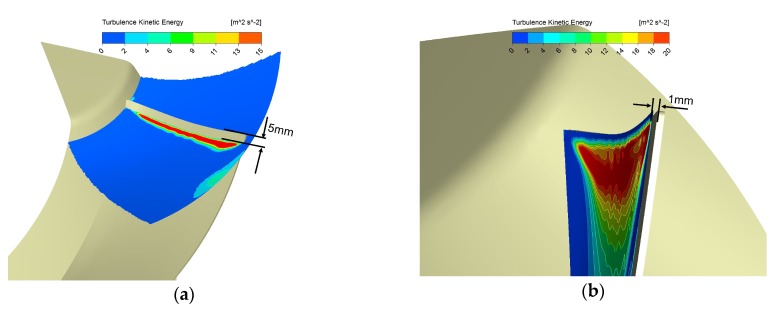
Locations for turbulence kinetic energy (TKE) quantifying and presenting: (**a**) Spanwise face, 5 mm from the leading edge chordwise (SF_5mm), (**b**) Offset face, 1 mm from the suction surface in its normal direction (Off_Suc_1mm).

**Figure 5 biomimetics-04-00072-f005:**
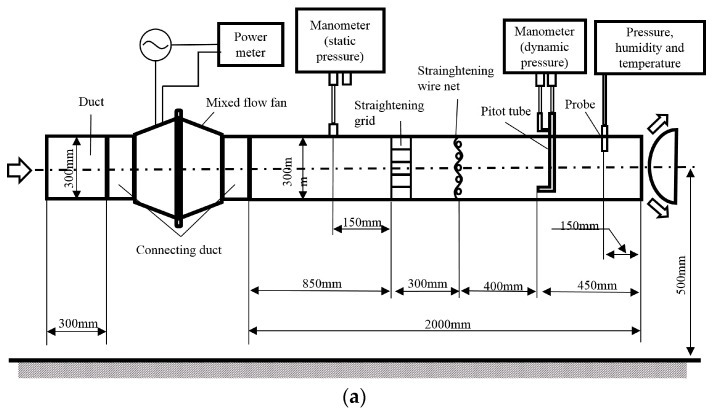
Validation of a 2-blade full-size model by results in our previous study [32]: (**a**) Experimental facilities, (**b**) Basic 2-blade full-size model, (**c**) Local details of the exaggerated region in (**b**), (**d**) Results of Total Pressure versus Flow Rate for the “Basic” and the “Serration L15A20” design derived from experiments (EXP) and simulations (SIM), (**e**) Results of Efficiency versus Flow Rate for the “Basic” and the “Serration L15A20” design derived from experiments (EXP) and simulations (SIM). The operating conditions of the serrations design in (**d**) and (**c**) and their nearby ‘basic ones’ are very close to that in Table 1.

**Figure 6 biomimetics-04-00072-f006:**
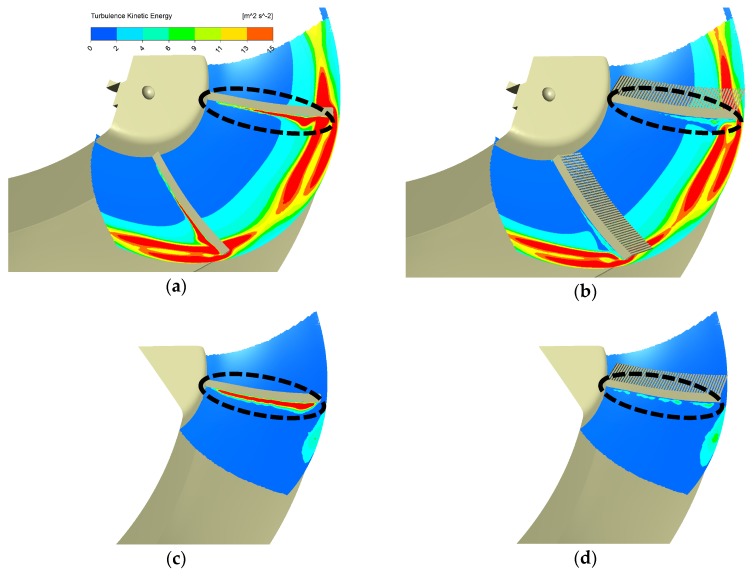
TKE contours on SF_5 mm of the 2-blade full-size model and the single blade model (the following values in brackets are TKE area integrals, unit: m^4^/s^2^): (**a**) Basic 2-blade full-size model (0.112), (**b**) Serrations’ 2-blade full-size model (0.0997), (**c**) Basic single blade model (0.00990), (**d**) Serrations’ single blade model (0.00406). Note that the reduction of TKE for each blade averagely derived from the 2-blade full-size model is 0.00615, while the value derived from the single blade model is 0.00584. The error is 0.00031, which is 5% of 0.00615.

**Figure 7 biomimetics-04-00072-f007:**
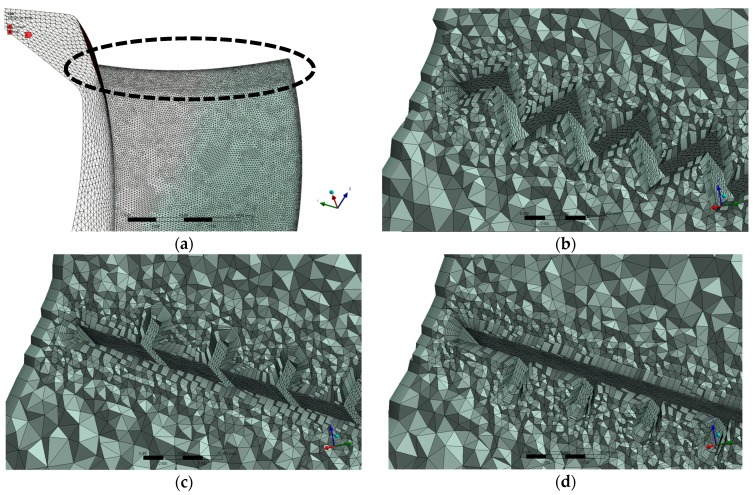
Grids of different blade forms: (**a**) Basic blade surface mesh, (**b**) Spanwise section mesh of the wavy blade, (**c**) Spanwise section mesh of the riblets on suction face blade, (**d**) Spanwise section mesh of the riblets on pressure face blade. The regions in the circle (leading-edge region) in (**a**) is refined with a grid size of 0.7 mm, which is applied to the other designs meshes as well.

**Figure 8 biomimetics-04-00072-f008:**
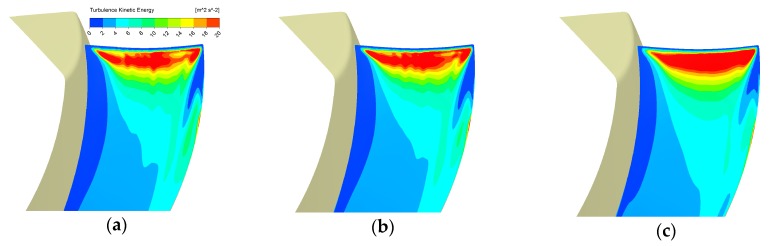
A verification of TKE results on Off_Suc_1 mm for different mesh sets: (**a**) Mesh set 1, (**b**) Mesh set 2, (**c**) Mesh set 3.

**Figure 9 biomimetics-04-00072-f009:**
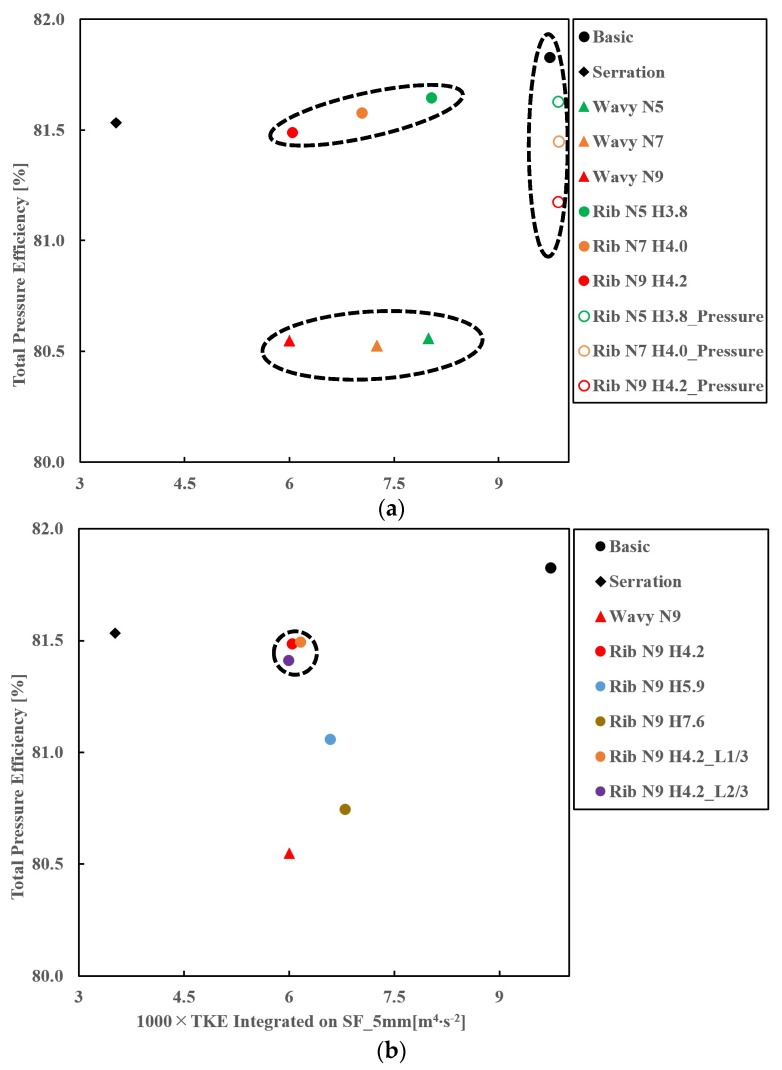
Total Pressure Efficiency versus 1000 × TKE of different blade designs. (**a**) Performance of different biomimetic design sets, (**b**) Exploration of groove parameters for the riblets on suction face designs. Note that the “Rib” set without “Pressure” represents riblets on suction face designs.

**Figure 10 biomimetics-04-00072-f010:**
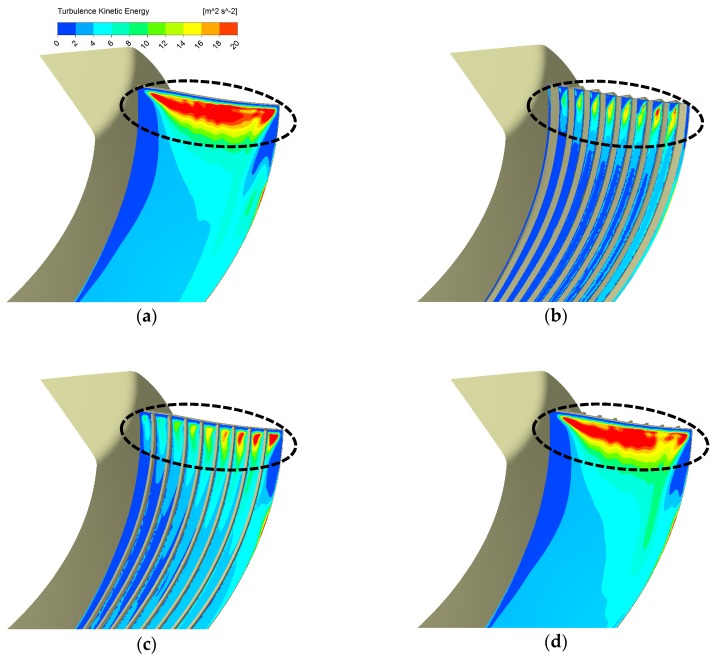
TKE contours on Off_Suc_1 mm of different blade forms: (**a**) Basic, (**b**) Wavy_N9_H3, (**c**) Rib_N9_H4.2, (**d**) Rib_N9_H4.2_Pressure.

**Figure 11 biomimetics-04-00072-f011:**
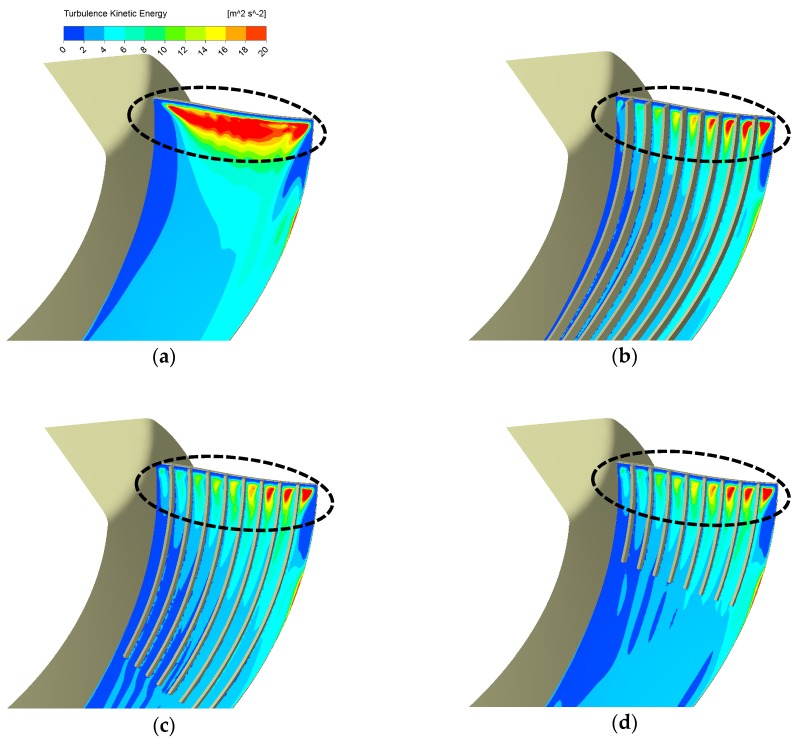
TKE contours on Off_Suc_1 mm of the basic design and various riblets on suction face designs: (**a**) Basic, (**b**) Rib_N9_H7.6, (**c**) Rib_N9_H4.2_L2/3, (**d**) Rib_N9_H4.2_L1/3.

**Figure 12 biomimetics-04-00072-f012:**
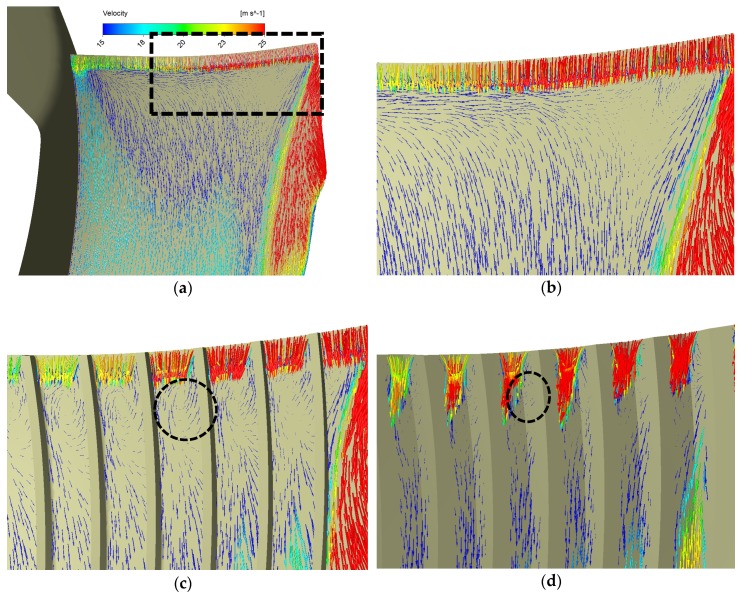
Velocity vectors (in rotational frame) projected on Off_Suc_1 mm: (**a**) Exaggerated region illustration, (**b**) Basic, (**c**) Rib N9 H4.2, (**d**) Wavy N9 H3. The rectangle region in (**a**) is exaggerated as in other figures and the circles denote the locations of vortices generated.

**Figure 13 biomimetics-04-00072-f013:**
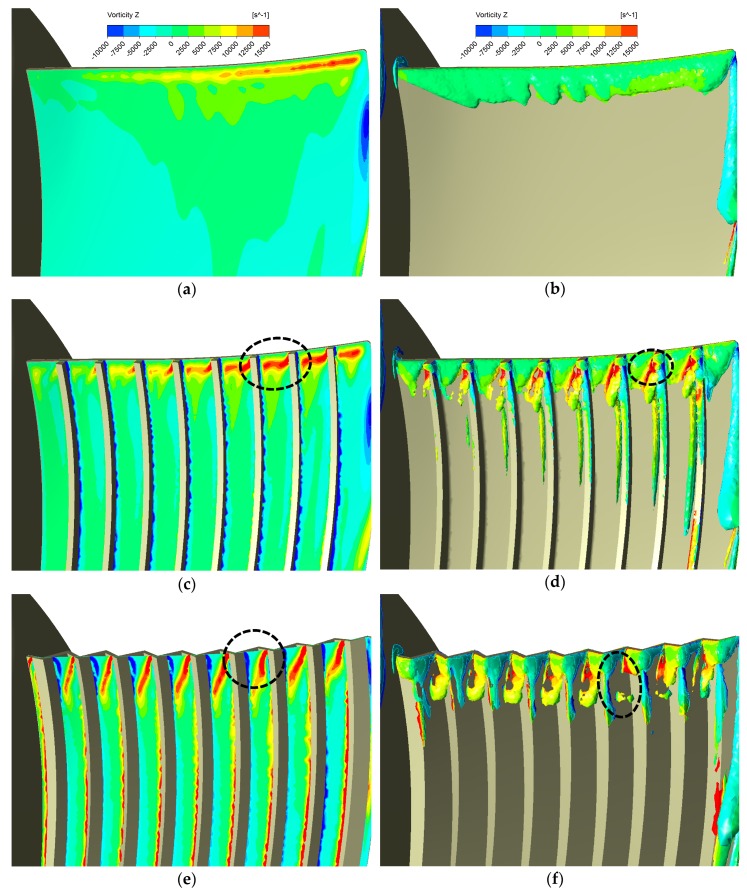
Axial vorticity on Off_Suc_1 mm and vortex core region (Q = 0.02): (**a**) and (**b**) Basic, (**c**) and (**d**) Rib N9 H4.2, (**e**) and (**f**) Wavy N9 H3. Regions in circles show obvious stronger axial vorticity than the basic.

**Table 1 biomimetics-04-00072-t001:** Parameters of the mixed flow fan.

Parameters	Values
Maximum Diameter of impeller/mm	426
Number of blades	6
Duct length/mm	412
Flow rate/m^3^/min	35.7
Rotational speed/rpm	1450

**Table 2 biomimetics-04-00072-t002:** Parameters of the biomimetic designs.

Parameters	Values
Groove number *N*	5, 7, 9
Wavy shape groove height H/mm	3
Riblet height H/mm (for the ones on the suction face)	3.8 (*N* = 5) 4.0 (*N* = 7) 4.2, 5.9, 7.6 (*N* = 9)
Riblets length proportion (for *N* = 9 and H = 4.2mm)	$\frac{1}{3},\frac{2}{3},1$

**Table 3 biomimetics-04-00072-t003:** Parameters and numerical results of three sets of mesh.

Parameters and Results	Mesh Set 1	Mesh Set 2	Mesh Set 3
Blade surface grid size/mm	0.7	0.7 (only leading-edge region)/1	1.5
Rotational domain body size/mm	3	4	5
Element number ($\times 10^{6}$)	6.40	2.63	1.26
y+ on blade (based on Re = 446,000)	5	10	25
Growth rate of prism layer (blade)	1.1	1.06	1.2
Prism layer number (blade)	15	11	5
Outlet total pressure/Pa	366	368 (+0.5%)^1^	374 (+2.2%)
Efficiency (×100%)	81.9	81.8 (−0.1 pp)	81.9 (0 pp)
1000× TKE Integrated on SF_5 mm/ m^4^/s^2^	9.77	9.74 (−0.3%)	11.30 (+15.7%)

^1^ The values in brackets are the errors compared with the Mesh Set 1.

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
