# Peer review of "Development of Mixed Flow Fans with Bio-Inspired Grooves"

_biomimetics, 2019, doi:10.3390/biomimetics4040072_

Round 1
Reviewer 1 Report
Development of Low Noise Mixed Flow Fans with Bio-inspired Grooves
Biomimetics 550724
In the present manuscript, the authors numerically investigated the effects of grooved and riblet blades on a mixed flow fan. It was mainly shown that such biomimetic designs have potentials in noise suppression. The topic of the manuscript is of interest and importance, but the current paper falls far short of presenting a robust numerical investigation. In overall, the authors could have done a lot better in presenting the contents of the manuscript, and particularly, in evaluating the computational methods and explaining the results. The manuscript also lacks adequate descriptions of methods and originality such that its contribution to the field would be trivial if any. I will cite a few, in case the authors want to revise the manuscript deeply and resubmit,
• The English of the paper needs improvement and should be articulated. The authors need to
proofread the manuscript and consider rewriting some parts of the manuscript.
• The introduction does not provide sufficient background, and definitely, does not include all
relevant references. There exists a reach history behind passive flow control techniques that are
almost entirely neglected by the authors.
• In overall, the numerical method representation is not acceptable in the current form. The numerical method should be evaluated by considering and resolving the following points,
o The description of the numerical method must be precise and include an assessment of the
order of accuracy of the truncation error introduced by each term in the governing equations. For instance, only stating that the simulations are based on the finite-volume formulations is definitely not enough.
o The numerical method used in solving turbulent flows must be at least second-order accurate in space for interior grids (the effect of numerical diffusion on the solution accuracy is usually devastating for first-order methods unless otherwise proved in detail).
o Solutions over a range of different grid resolutions should be presented to demonstrate grid-independent or grid-convergent results. Besides, the use of error estimates based on methods such as Richardson extrapolation may also be employed to illustrate solution accuracy.
o Stopping (convergence) criteria need to be precisely explained for iterative calculations.
o The boundary and initial conditions must be clearly presented along with the methods used to implement them. Moreover, in simulating freestream turbulent flows, particular attention should be paid to the treatment of inflow and outflow boundary conditions.
o The governing equations that are solved to obtain the presented results need to be stated.
o Comparison with reliable experimental results is required, provided experimental uncertainty is established.
o The computational setup, including the grid, should be presented in the paper along with a close view of grid points inside the grooves and riblets. Moreover, the grid information is not complete. The authors need to report the height of the first grid row, the growth factor, and the peak Y-Plus value.
• The explanations behind the turbulence model selection are inadequate. Furthermore, the use of a RANS model is not justified in the paper; which scales are modeled and why can they be modeled should be explained.
• One of the critical flow features in aerodynamics performance is to consider the amount of laminar run of flow. This cannot be ignored without detailed justifications. The authors can support their fully turbulent flow assumption by the works of, for example, Refs. [1-2].
• The flow includes separation which is sensitive to unsteadiness, so actually, it is an unsteady problem. The authors need to explain this assumption?
• The manuscript focuses on the application of turbomachinery, and in particular, mixed flow fans. However, I seriously doubt it to be noteworthy without consideration of blades interaction.
• How the authors choose the height of the grooves and riblets? The groove's geometric characteristics should be selected on a physical basis rather than on an arbitrary basis. The height (or what I call depth) of grooves should be proportional to the boundary layer thickness. Moreover, instead of keeping the area between grooves and riblets fixed, it is more reasonable to keep the ratio of the boundary layer thickness to the (groove/riblet) height fixed.
• The manuscript lacks physical discussion, and the authors fail to provide an explanation of the flow control effectiveness in terms of physical understandings. There are other comments (far too many to list here) requiring adept editing, but I stop here. If the editor agrees for a resubmission, then I will see how much of the above corrections are assimilated by the authors.
References
[1] Yousefi, K., & Razeghi, A. (2018). Determination of the Critical Reynolds Number for Flow Over Symmetric NACA Airfoils. In Proceedings of the 2018 AIAA Aerospace Sciences Meeting, AIAA 2018–0818, Kissimmee, FL, USA.
[2] McMaster, J. H., & Henderson M. L. (1979). Low-speed single-element airfoil synthesis. NASA
Technical Reports, 19790015719.
Author Response
Please see the attachment. The cover letter and replies follow the revised manuscript.

Reviewer 2 Report
The paper can be accepted after some following minor revisions:
1. The manuscript should give more details of the CFD. The efficiency is between 80% and 81.5% from figure 5 and 7 of different blade form set. So, the accuracy of numerical simulation n is particularly important.
2. Please provide descriptions, such as experimental verification, to prove that your optimization results are correct.
3. The relationship between noise and TKE should investigate more because the noise source is not just decided by the wall pressure. Turbulent noise could be dominate in some cases.
4. TEK should be changed to TKE in line 119.
5. Conclusions put too much and need to be condensed into 3 or 4 pieces. Moreover, it is inferred that the optimization of noise is not yet mature from the conclusion of conclusion 7, and it is suggested that the title be changed without noise.
Author Response

(The authors gave the same response as above.)

Round 2
Reviewer 1 Report
See the attached file.

Author Response
Please see the attachment.
Location changes of figures are not recoded by "Track Changes" function in Word.

Round 3
Reviewer 1 Report
The paper can be accepted. There are just two minor modifications.
Line 125: Introduce U_i and k in equations 4 and 5.
Line 224: Change "Hence, the flow condition over the blade is almost a total fully turbulent flow as the transition happens pretty close to the leading edge" to "Hence, the flow condition over the blade is fully turbulent, and the transition happens pretty close to the leading edge."
The English still needs some final polishes.
Author Response
The paper can be accepted. There are just two minor modifications.
Line 125: Introduce U_i and k in equations 4 and 5.
Answer:
We added the descriptions of the two terms in the manuscript accordingly (Line 106-108).
Line 224: Change "Hence, the flow condition over the blade is almost a total fully turbulent flow as the transition happens pretty close to the leading edge" to "Hence, the flow condition over the blade is fully turbulent, and the transition happens pretty close to the leading edge."
Answer:
This is fixed (Line 200).
The English still needs some final polishes.
Answer:
We doubled check the manuscript and made revisions to make the manuscript more readable.
PS:
The captions in Figure 9 (a) & (b) have been revised from the “Wave N_” to “Wavy N_” to make them consistent with the expressions in the manuscript; We revised a couple of places to make it clearer that the generated vortices by grooves are not necessarily axial though they definitely own the axial components and the axial Z vorticity was checked merely for convenience. (Line 356-357, Line 402)